civil engineering

resilience, tornado, recovery, modelling

**Author for correspondence:**
Stephanie F. Pilkington
e-mail: spilking@uncc.edu

# Update article: applicability of artificial neural networks to integrate socio-technical drivers of buildings recovery following extreme wind events

## Stephanie F. Pilkington[1] and Hussam Mahmoud[2]

[1]Department of Engineering Technology and Construction Management, University of North Carolina at Charlotte, 9201 University City Boulevard, Charlotte, NC 28223, USA
[2]Department of Civil and Environmental Engineering, Colorado State University, 1372 Campus Delivery, Fort Collins, CO 80523, USA

HM, 0000-0002-3106-6067

In a companion article, previously published in *Royal Society Open Science*, the authors used graph theory to evaluate artificial neural network models for potential social and building variables interactions contributing to building wind damage. The results promisingly highlighted the importance of social variables in modelling damage as opposed to the traditional approach of solely considering the physical characteristics of a building. Within this update article, the same methods are used to evaluate two different artificial neural networks for modelling building repair and/or rebuild (recovery) time. By contrast to the damage models, the recovery models (RMs) consider (A) primarily social variables and then (B) introduce structural variables. These two models are then evaluated using centrality and shortest path concepts of graph theory as well as validated against data from the 2011 Joplin tornado. The results of this analysis do not show the same distinctions as were found in the analysis of the damage models from the companion article. The overarching lack of discernible and consistent differences in the RMs suggests that social variables that drive damage are not necessarily contributors to recovery. The differences also serve to reinforce that machine learning methods are best used when the contributing variables are already well understood.

# 1. Introduction

With the growing interest in applying machine learning methods to modelling various real-world problems, a research article previously published by the authors evaluated the use of graph theory to potentially interpret the black box of artificial neural network (ANN) models [1]. This analysis was conducted with the goal of better understanding socio-technical patterns constructed in resulting ANN models that attribute to wind damage following an extreme event. Graph theory concepts, including centrality and shortest path, were originally used to define the importance of a neuron, or variable, as well as highlight how that variable may contribute to an overall resulting building damage state. The results showed that when social variables were included in the model, the network appeared more organized (determined from a centrality analysis), and a shift in input variable importance occurred to focus more on *how* the building was being used instead of its primary structural characteristics (determined from a shortest path analysis). These preliminary findings consider the prospect that the use of graph theory may be viable in understanding complex problems modelled using ANNs or other machine learning methods.

As the conclusion of the original article highlights [1], further studies with different ANN models would provide additional insight into the applicability of graph theory to analyse ANNs and, perhaps, the applicability in using ANNs for modelling purposes in general. As such, the authors built various new ANN models for evaluation through this method. However, these models are intended to predict building recovery over time, as opposed to damage, from extreme wind hazard events. The original article in evaluating damage constitutes the initial impact phase. To model resilience, using machine learning methods, the time required to rebuild damaged structures (recovery) must also be considered. While there are physics-based approaches to modelling damage, i.e. evaluating the probability of failure for a specified wind speed, modelling the recovery process is still relatively new. A significant understanding between the hazard event, the infrastructure being rebuilt and the social aspects of the community doing the rebuilding are still being researched. This makes modelling recovery highly challenging. As a result, applications such as ANNs would seem appealing to model community recovery from extreme events.

There are published preliminary models of recovery for community resilience purposes. In 2011, the ResilUS model was introduced to represent damage and recovery over time of a community's capital [2]. The 'capital' considered within this model was represented through three aspects: the built environment, economics and social. Each of these was influenced by variables that not only included the hazard but the building types, probabilistic availability of resources (materials and monetary), building occupant's loans and debt, time to file and receive resources (loans) and even injuries to building occupant/owner [2]. Within the modelling software itself (Matlab), Markov chains were used to model recovery over time. However, this model did not include the use of GIS software for explicit spatial distribution of community-wide recovery.

Another potential model using Markov chain analysis evaluated the recovery time of a building stock, in which each building was evaluated for how long it would take to reach the 'fully functional' state [3]. According to Lin & Wang [3], there are two phases considered in determining a recovery time: (i) delay time, which includes inspection and permitting, and (ii) rebuilding. In modelling such recovery time, Lin & Wang [3] used Markov chain analysis to include the uncertainties in the post-disaster functionality state, and decisions owners may make that influence the delay time [3]. Recent development in recovery modelling also includes dynamic finite-element analysis of resilience model structure, where it was proposed that a community can be divided into grids and treated as an area to be meshed similar to how finite-element analysis is executed on structural components following the general dynamic equation of motion [4]. Other methods that can be used to model recovery include empirical methods and economic theories, among others. The ANN models discussed herein use hazard, structural and social data to determine the recovery time of a structure following an extreme wind event. Recovery, for the purposes of this article, refers to the time required for a structure to be repaired or rebuilt and reoccupied such that it has returned to its original functionality state. The same methods as outlined in the previous companion article were used to build and select the ANNs for evaluation as well as the analysis approach with graph theory. The main difference in methodology concerns the collection of data relevant to building a recovery ANN model and will be discussed herein.

# 2. Data collection for recovery

The same data points used in constructing the original damage model ANNs, which consist of social and structural input variables corresponding to extreme wind events in the state of Missouri from 2011 to

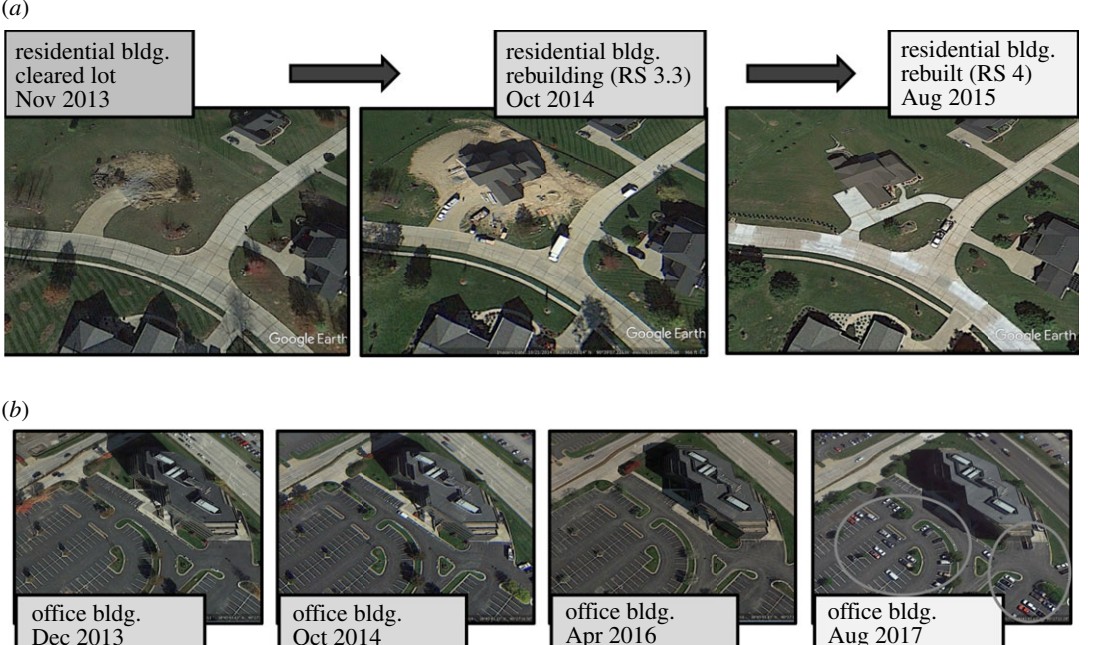

**Figure 1.** Example Google Earth images and corresponding recovery states over time for (*a*) DS2 residential building that decided to rebuild from 1 June 2013, tornado event and (*b*) DS1 office building (windows were blown out) that was not reoccupied until August 2017 following 1 June 2013, tornado event (circles show indications of building in use).

2015, were used for the recovery model (RM) dataset. As such, the source of social demographic information remains the US Census [5] with historical event post-storm survey image data collected from the US National Weather Service (NWS) [6,7] for structural assessments. The missing data required to build recovery ANN models were, therefore, the output data that communicate time to return to pre-event functionality.

Determining recovery involves evaluating the changing state of a data point (structure) through time. Once a structure from the original dataset was geographically located within Google Earth based on the NWS's geo-tagged photos, a visual assessment was made of the structure prior to the hazard event and up to 3 years following the event, if possible. Each structure's condition was assessed for whether it was rebuilt and reoccupied by six months, 1 year, 1.5 years, 2 years or not yet recovered by 2 years. Each structure (feature) was given a recovery score at each time step. A building was considered 'good as new' when it reached a recovery state 4. The output data for the ANNs were coded as whether it had recovered six months, 1 year, 1.5 years, 2 years or not yet recovered by 2 years.

As with analysing the original damage-state photos, there were some complications in evaluating satellite images. The main issues were with concern to where a photo was geo-located and identifying the building associated with the photo within Google Earth. In these cases, neighbouring features presented in both the photo and satellite imagery were compared to determine which structure was being evaluated. There were instances where a structure could not be either located or evaluated from the satellite imagery. These cases were classified as 'ND' or 'no data' and were removed from the dataset, resulting in a final tally of 93 data points available for building the RMs. Other instances causing an ND classification included: satellite images with a large time span gap (example: image availability only in 2012 and 2015) with no way to ascertain when the structure was repaired, and images that may still show a structure damaged or being repaired, but there are no more recent satellite images to determine when repairs were completed. A damaged structure, as it appears through Google Earth at various points in time, is shown in figure 1. Figure 1*a* shows a standard progression in rebuild, while figure 1*b* shows how occupancy was also used in determining recovery for buildings with damage states that were less obvious from a satellite view, such as Damage States 1 and 2.

# 3. Artificial neural network structure

There are two building recovery ANN models primarily discussed herein, one for (A) considering only social vulnerability factors (Recovery Model 2, RM2) and one for (B) a mix of hazard characteristics,

**Table 1.** List of variables for damage and recovery models. The models used in the analysis are indicated in bold text.

| input variable | associated recovery models |
| --- | --- |
| hazard: (wind type, speed, and event size) | 1 **8** 9 10 11 14 |
| structural: (year built, occupancy, roof and wall materials, roof shape, footprint) | 1 4 5 6 7 **8** 9 10 11 12 14 |
| surface roughness | 1 **2** 3 4 **8** 9 10 11 12 14 |
| estimated per cent forested: (and impervious surfaces) | 1 **2** 3 4 **8** 9 10 11 12 14 |
| tenure: (% own, % rent, single female head of household w children, group quarters) | 1 **2** 3 4 5 6 7 **8** 9 11 12 15 |
| housing and population density | 1 **2** 3 4 5 7 **8** 9 10 11 12 14 |
| total population | 1 **2** 3 4 5 6 7 **8** 9 10 11 12 14 |
| age | 1 **2** 3 4 5 6 7 **8** 10 11 12 14 |
| race: (% Asian, African American, Native American and Hispanic) | 1 **2** 3 4 5 6 7 9 10 11 |
| industry employment: (extractive and service) | 1 **2** 3 4 5 6 **8** 9 10 11 12 14 |
| income: (per capita and income: poverty) | 1 **2** 3 4 5 6 7 **8** 9 10 11 12 14 15 |
| disability | 1 **2** 3 4 5 6 7 **8** 9 10 11 12 14 15 |
| persons over 65 years old | 1 **2** 3 4 5 6 7 **8** 9 10 11 12 14 15 |
| no vehicle | 1 **2** 3 4 5 6 7 **8** 9 10 11 12 14 |
| residence for at least 1 year | 1 **2** 3 4 5 6 7 **8** 9 10 12 14 15 |
| damage state | 1 **2** 3 5 6 7 **8** 9 10 11 14 15 |

structural characteristics and social factors (Recovery Model 8, RM8). These two models were selected for comparison out of 15 possible RMs. The primary difference in these 15 models concerns the input variables. The relationship between the input variables and the models is shown in table 1. These models were evaluated for 'performance characterizing indicators (PCIs)' following the same analysis procedure as outlined in the original article's supplementary information [1]. Recovery was considered to rely more heavily on the social variables, as opposed to modelling damage, which was considered to rely more heavily on structural characteristics. As such, the initial 15 models contain a wide array of potential social variables contributing to recovery identified from various social vulnerability studies [8–13], which is similar to that which was considered in the initial model for damage predictions [1]. The outputs to these models were classified by building recovery state as described in table 2. While Model B (RM8) performed the best in establishing patterns between the possible variables and recovery time, Model A (RM2) produced slightly less desirable PCIs (figure 2). This would suggest that building materials (perhaps availability and ease of construction) do factor into the recovery time.

As with the original damage models, the ANNs consist of a feed-forward structure with a single hidden layer of 10 neurons and are trained using Bayesian methods. Error was also measured using mean square error. The final RMs, used for further analysis with graph theory, consist of an ensemble of multiple, specifically six, ANNs evaluating the same input data simultaneously.

## 4. Results in analysing ANNs using graph theory

The graphical analyses of the RMs were conducted in the same manner as the damage models [1]. This included creating two options for evaluating models consisting of six ANNs: (i) combining the six ANNs into one graphical network and (ii) evaluating each ANN individually and averaging the results. In the shortest path analysis, if a relatively strong connection (high weight values) was found in both the combined and averaged approach, then that was considered a finding of this analysis. Similarly, the weaker connections were also evaluated. Figure 3 presents the relative results from calculating the shortest path for Model A (RM2) with the primarily social inputs model structure. Note that certain structural parameters, such as size and occupancy, were kept as inputs in RM2, as these were

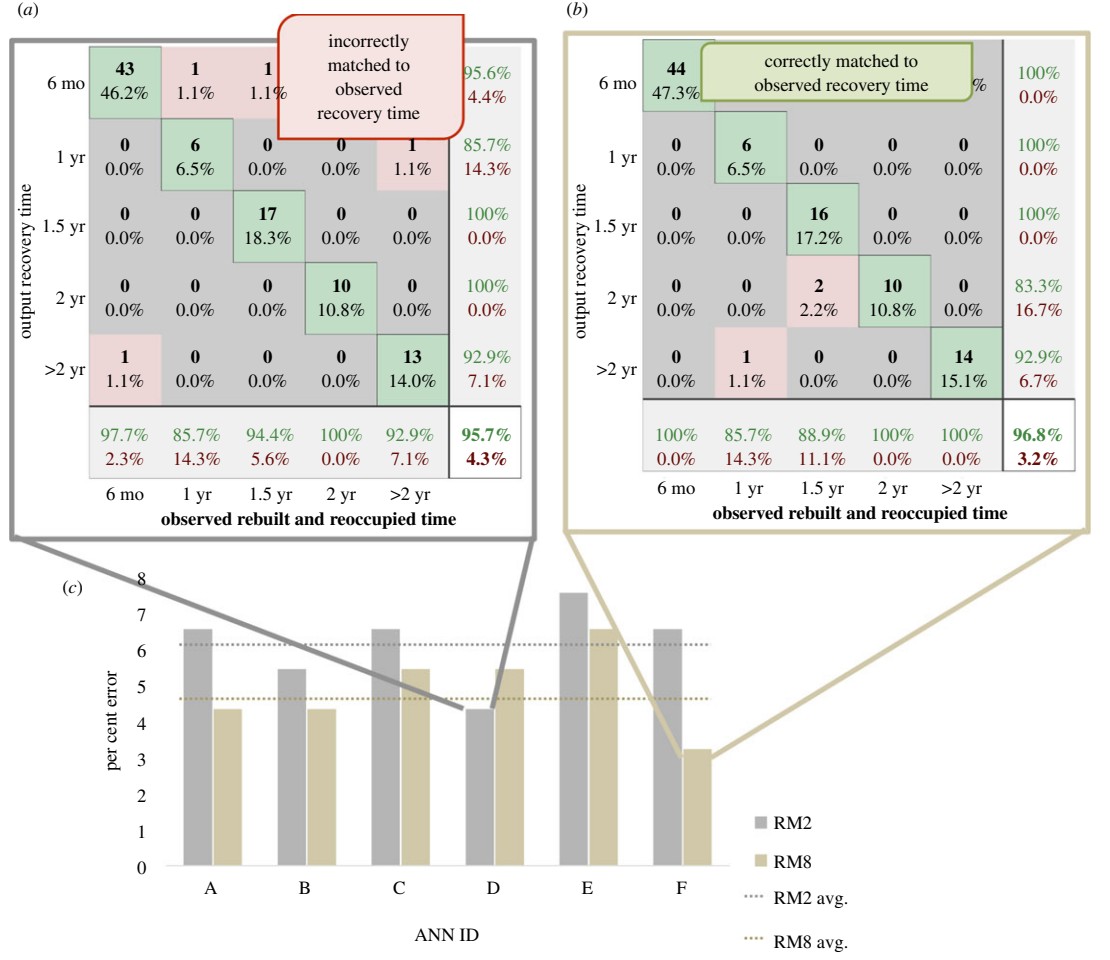

**Figure 2.** Comparison of Model A (RM2) and Model B's (RM8) performance in building resulting ANNs.

**Table 2.** Recovery states defined by Curtis & Fagan [14].

| recovery state | description | sub-category | elaboration |
|---|---|---|---|
| 1 | uninhabited | 2 | liveable: unoccupied |
| | | 5 | blighted |
| | | 10 | non-liveable: extreme |
| 2 | cleared | | lot empty due to destroyed home or clear for reconstruction |
| 3 | rebuilding | 1 | Frame skeleton is up. This would only appear for homes needing a complete rebuild. |
| | | 2 | walls are enclosed |
| | | 3 | Non-structural components have been added. It is likely that DS2 and DS3 would not require more than this. |
| | | 4 | cosmetic finishes |
| 4 | rebuilt and occupied | | 'good as new' |
| 5 | no rebuild/new structure | | abandoned lot |

originally considered foundational variables for reconstruction. In RM2, the building size, in terms of footprint area, was linked to a six-month recovery time, whereas building height was found to strongly link to a 1-year recovery time. Additionally, building damage state was found to strongly

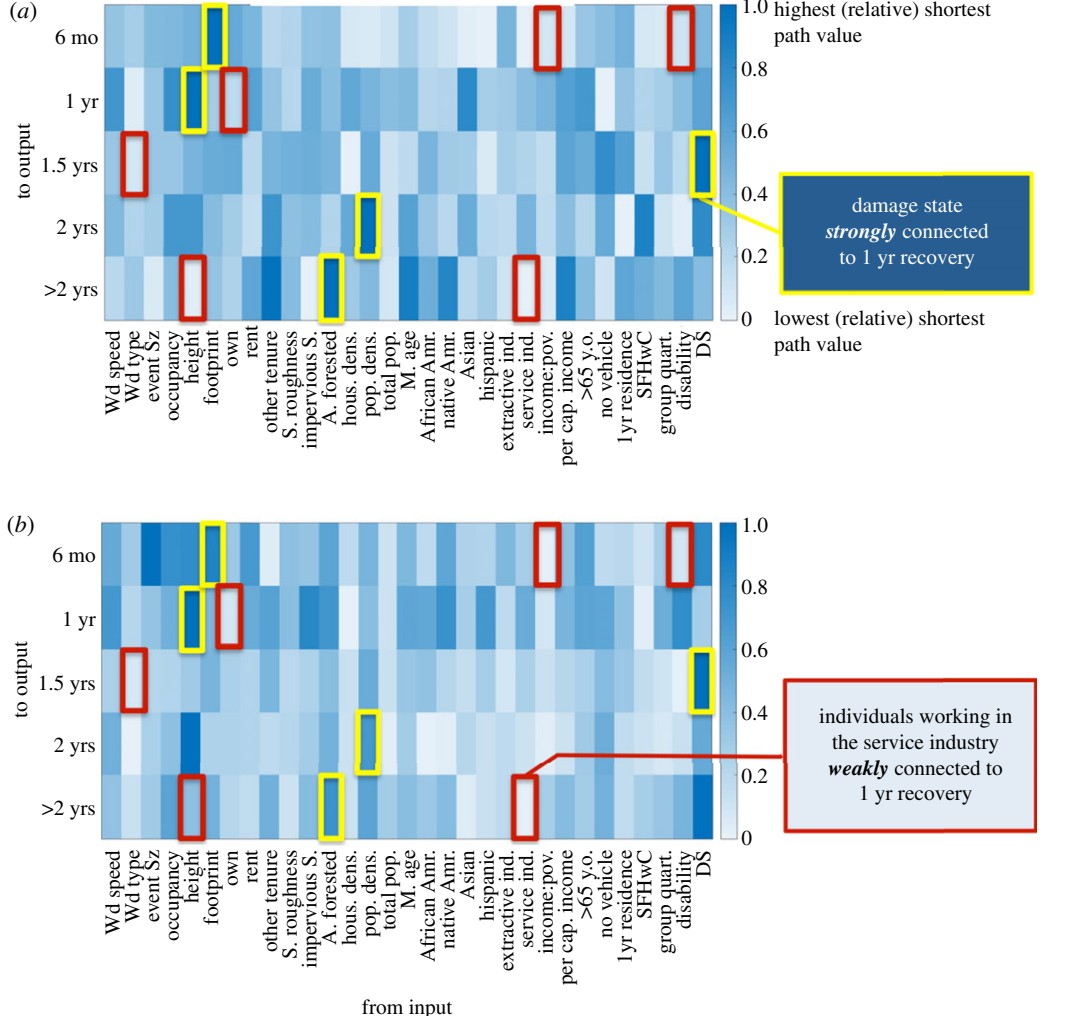

**Figure 3.** Model A (RM2) shortest path relative values for the (*a*) combined ANNs and (*b*) averaged results from each ANN approaches.

connect to a 1.5-year recovery time, population density to 2 years and area forested to the building taking longer than 2 years to recover (or become abandoned). Weaker connections were found between the amount of people on disability and the income to poverty ratio to a six-month recovery time. Owning tenure and the wind event type (straight line or tornadic) were shown to be weakly connected to 1- and 1.5-year recovery times, respectively. Both building height and the amount of people working in the service industry demonstrated a weak connection to determining if the structure would be abandoned in RM2.

The results from analysing RM2 provide an initial description of the network connection structure without parameters related to building materials. Figure 4 presents the shortest path analysis results after introducing building variables and subtracting out racial variables in the ANN structure to form Model B (RM8). While building height still ties strongly to a 1-year recovery time, it has also tied strongly to the six-month recovery time, which is more intuitive since building height tied strongly to no damage (DS0) in the previous graphical analysis for the damage models [1]. The wind speed and a single female head of household with children were also found to strongly connect to a 1-year recovery time, while the roof shape was strongly connected to an over 2-year recovery or abandonment. Conversely, individuals with disability moved from having a relatively weak connection to six-month to a 1-year recovery time. The structure's median year built and occupants working within extractive industries were found to weakly connect to recovery by 1.5 years. While building height remained strongly connected to earlier recovery times and disability weakly connected to earlier recovery times, the remaining strong and weak connections did not suggest any significant shifts in focus. In dropping strong connections from building footprint, damage state, population density and area forested, strong connections were gained in relation to wind speed, roof

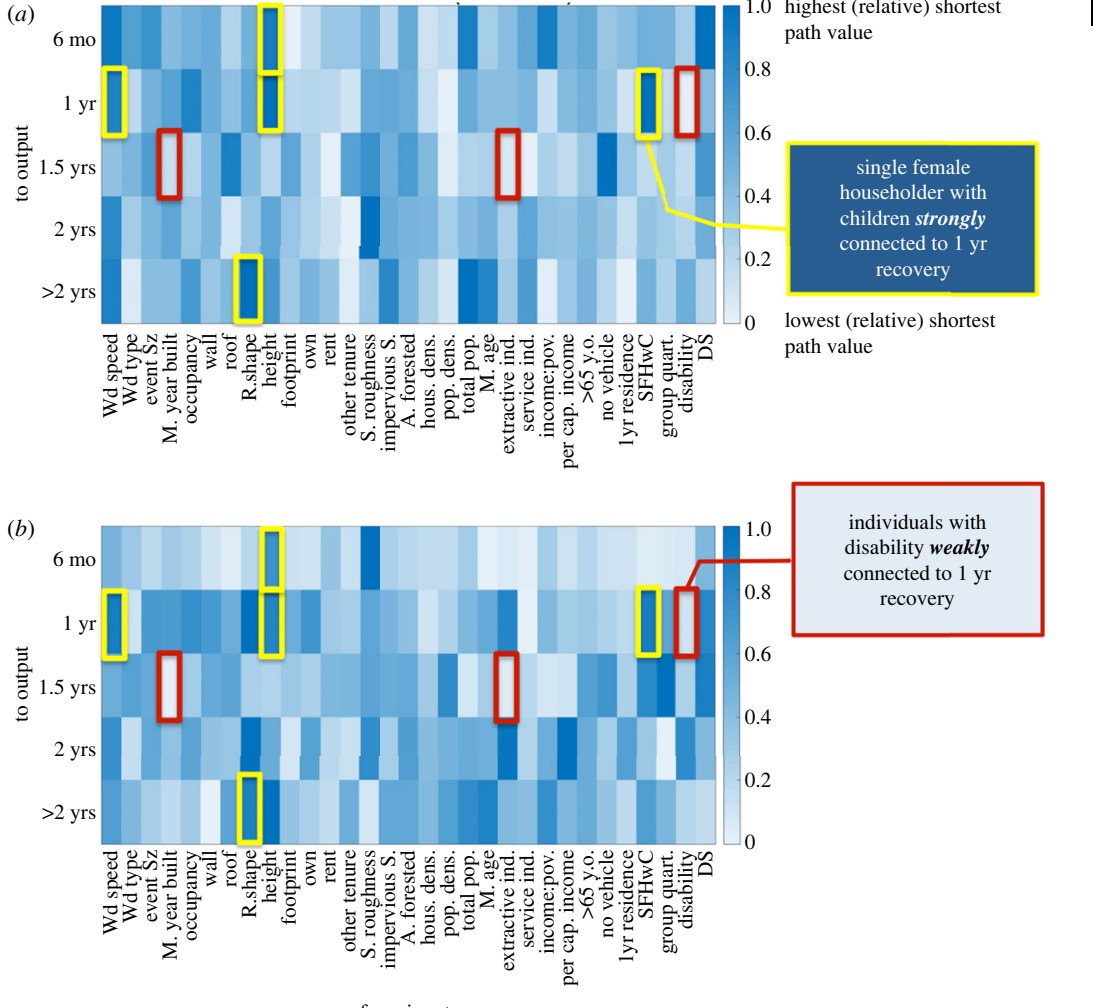

**Figure 4.** Shortest path relative values for RM8 (*a*) combined ANNs and (*b*) averaged results from each ANN.

shape and single female head of household with children for Model B. Model A's stronger connections focused more on the area of and surrounding the building, but the inclusion of structural variables does not seem to have replaced those connections with any similar pattern. A similar assessment could be said for the weaker connections, except for the shift from per cent of the population employed in service industries to per cent employed in extractive industries having weak connections.

As with the shortest path analysis, the centrality analysis, through the concepts of closeness and degree, was conducted for recovery modelling in the same manner as the damage model analysis [1]. Figure 4 shows the centrality scores, plotted as closeness versus degree. Model A resulted in a combined network where the recovery times of 1 to greater than 2 years were widely influenced by the multiple input variables. The lack of vehicle, other tenure (neither rent nor own), damage state, employment in extractive industries, income to poverty ratio, renting tenure, building occupancy code and footprint area are also widely connected within the network structure. In combination with the shortest path results, this would suggest that building footprint area and damage state are significant factors in determining recovery time within the Model A structure. Conversely, owning tenure is a less significant factor in this structure as it shows a weak connection to a 1-year recovery time and low centrality scores when related to the overall network.

Model B's centrality scores are shown in figure 5. Single female head of household with children, wind speed and per cent of area consisting of impervious surfaces were all inputs found to be widely connected to the possible recovery times within this network. The 1-year recovery time was also shown to remain a heavily influenced output, indicating that many variables contributed to determining this outcome, as opposed to the six-month recovery time, which showed a lower degree centrality. In combination with the shortest path results, the wind speed and single female head of

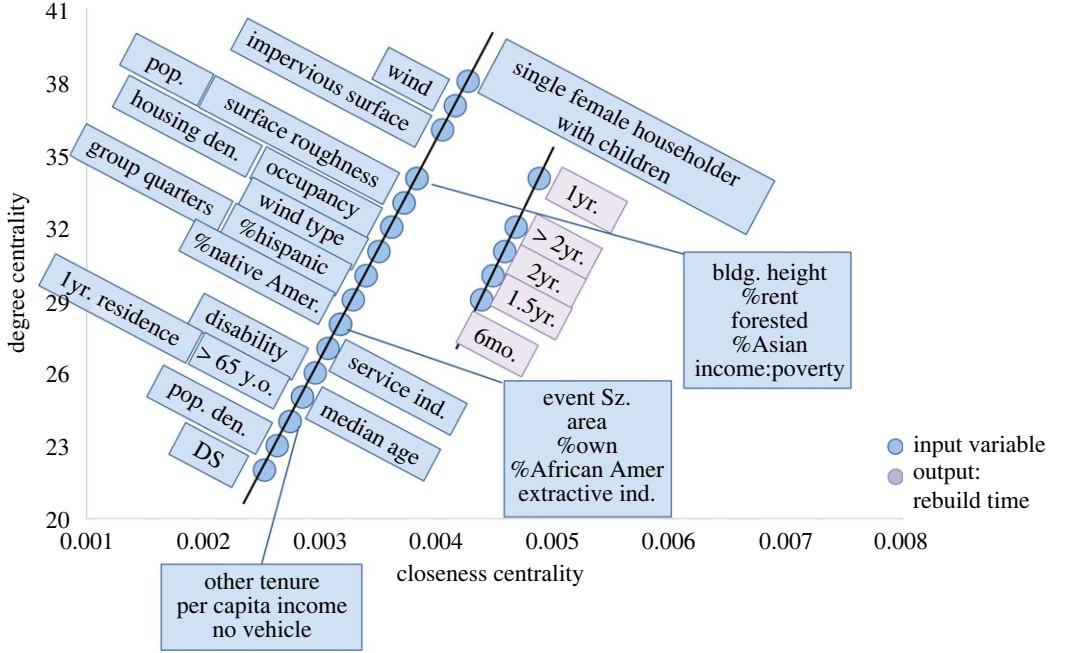

**Figure 5.** Model B (combined ANN structure) centrality scores as closeness versus degree.

household with children variables were found to be overall significant contributors to the time it would take to rebuild and reoccupy a structure following a severe wind event. In RM8, the variables that showed weak connections to specific recovery times did not also rank among the lowest centrality scores. This would indicate that while the population employed in extractive industries weakly tied to a 1.5-year recovery time, it also tied relatively well within the entire network overall.

The results from the centrality analysis showed similar organization among both input variables and output recovery times for both models, indicating that overall connectivity within the network may be similar even with a switched variable hierarchy and differing inputs. Additionally, the shortest path analysis results indicated a remaining importance of wind speed, building height and roof shape extending from modelling damage state to recovery. Overall, in comparison with modelling damage state, modelling recovery time proved to be a more involved and complex problem due to the overall higher per cent errors in building the ANNs and similarities in network organization between Model A and Model B through shortest path and centrality analyses. The lack of a significant shift in strong network connections and any organizational differences among the centrality plots brings into question how the ANN is built, given the data provided.

## 5. Model validation results

To further validate this approach as a method of predicting the time to rebuild and reoccupy a building, the 2011 Joplin, MO, tornado was simulated for Models A and B and then compared with documented recovery of select buildings from video data provided by Kent State University [14]. The tornado path and intensity were overlaid with the buildings in the area, similar to how the damage hindcast was conducted to include the wind hazard, structural type and US Census social characteristics.

As was performed with the damage models [1], the RM errors were assessed on a broad scale and a more exact scale. Each building within the video dataset was analysed for whether it was fully recovered (recovery state 4) by 1 year, 1.5 years, 2 years or longer. The six-month recovery data were not available within the actual Joplin video data; therefore, the ANN results would subsume six-month recovery under the 1-year recovery categorization. For the first error assessment, an exact match for each building was assessed for 1-year, 1.5-year, 2-year and greater than 2-year recovery times. For the second analysis, a plus or minus six-month error was introduced, such that if the ANN model predicted a 1-year recovery and a building actually reached recovery state 4 by 1.5 years, this was considered a match. This approximated match was used based on the assumption that if a singular building recovered by 10 months or 14 months, the overall impact to the community from such a difference would be

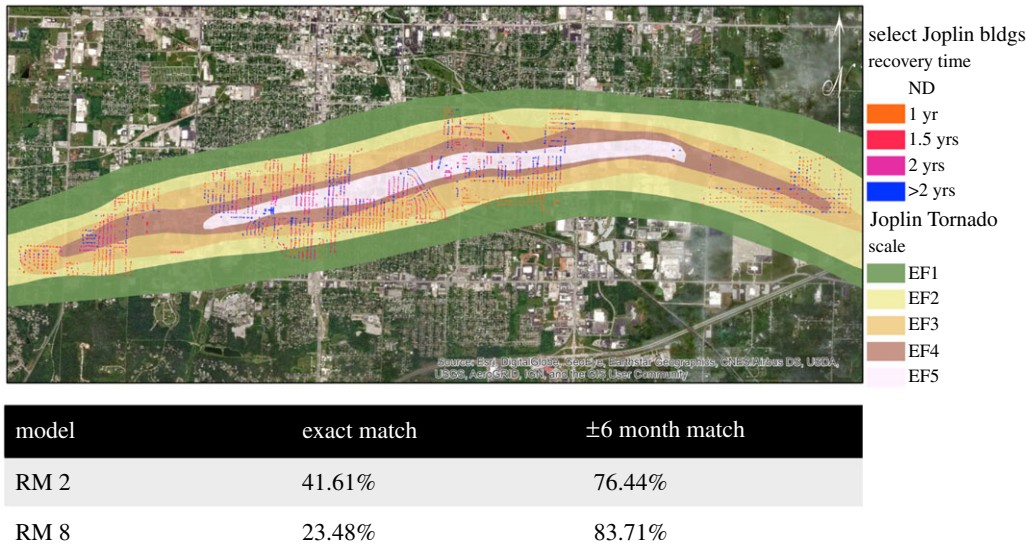

| model | exact match | ±6 month match |
|-------|-------------|----------------|
| RM 2 | 41.61% | 76.44% |
| RM 8 | 23.48% | 83.71% |

**Figure 6.** Validation results of Models A and B (RM2 and RM8, respectively) considering the 2011 Joplin tornado.

minimal. The results are shown in figure 6 with a match per cent (accuracy) of 23–42% for the exact match and 76–84% for the approximated match.

The recovery modelling errors proved to be higher than that of the damage modelling analysis. This was expected since there were differing damage-state categorizations across the multitude of post-event assessment surveys, which led to an error within the known dataset, and potentially greater uncertainties in categorizing a building's recovery state over time. Additionally, while consistency was maintained in conducting comparative performance assessment for the exact and approximate match approaches, Model A resulted in a lower error for the exact match when compared with Model B but switched for the approximated match.

The results of this additional validation using the 2011 Joplin tornado as a case study further reinforce the findings from the graphical analysis. The similar network organization (centrality analysis), lack of significant shift in neuron connections (shortest path) and the alternating of model accuracy (Joplin validation) bring to question the validity of using ANNs for modelling recovery.

## 6. Discussion

Of the 15 RM networks, the two evaluated (Models A and B) did include area forested, housing density and tenure, as were also included in the best performing damage model [1]. However, the differences between these two from the analysis results show more in common than different. The analysis of the damage models resulted in more apparent differences within the ANN building process, the graphical analysis and the hindcast. Models A and B mostly showed similar resulting network structures, with both networks showing organization (figures 4 and 7). The shift seen in the damage models once social factors were added was not seen in the RMs once structural characteristics (building materials) were added. Additionally, while both the damage model and RM showed negligible differences in hindcasting error, the RMs also switched which model performed slightly better for the exact match and approximated error. Accordingly, both Models A and B produced acceptable errors when considering a plus or minus six-month approximation and could both be considered viable model structures for this ANN application.

The lack of any significant difference between RMs in the overall analysis could also indicate missing variables. Either a specific connection type was not captured in the 15 model options or there were other factors to consider that have not yet been highlighted in the literature. The original 15 potential RMs did not include a specific parameter related to time to obtain building permits and set up construction jobs. The assumption that preceded creating these models was that some social characteristics may inherently overlap with these factors, such as low income or occupation, by tying an ability to manage the system and obtain a permit within a reasonable time frame to a general delay time characterized by social

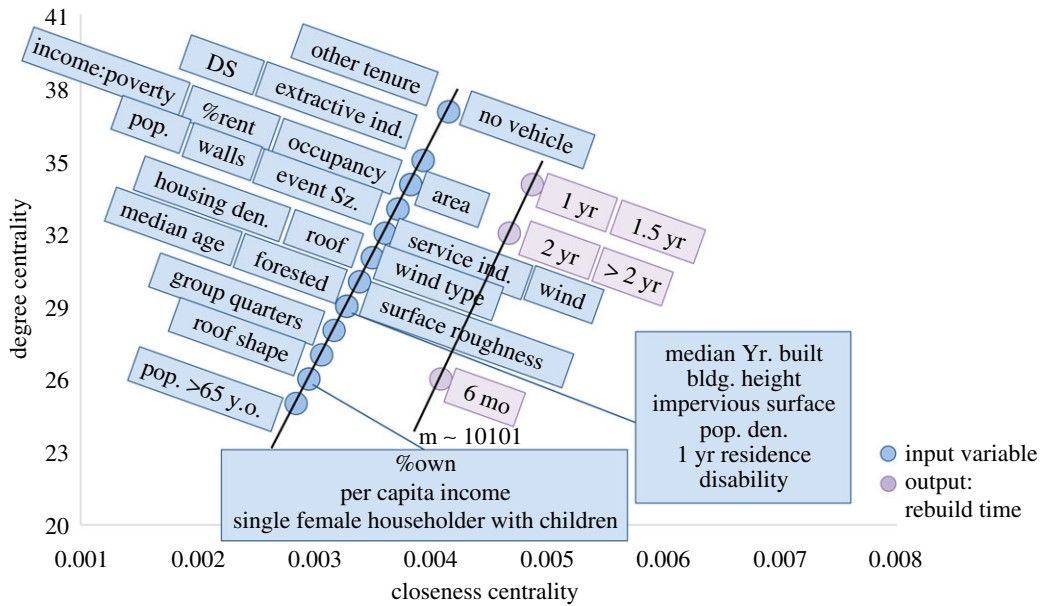

**Figure 7.** Model A (combined ANN structure) centrality scores as closeness versus degree.

demographics. However, no variables were included for the policy system itself, which could be considered an oversight within this research.

It is also plausible that there are factors that contributed to recovery in past events that cannot yet be captured with a numerical data point. For example, both Models A and B included the '1 year residency' variable as an attempt to show ties to a community. There is a concept of community mentality that could significantly affect how long it would take a community to recover. How close together (socially) is the community? Do they offer to help each other? Is their outlook following an event more accepting or defeatist? These parameters could arguably be more important than any other variables considered herein but are not easily quantifiable for the purposes of modelling with ANNs.

The potential importance of abstract social concepts, along with the lack of consistent differences between the evaluated RMs discussed herein, highlighted the potential low significance of structural characteristics to the recovery process. These characteristics were vital in modelling damage state or initial impact but did not appear as critical to modelling recovery time. Within the terms of resilience, this further highlights the importance of engineering variables primarily at the initial impact stage, while the social characteristics are integrated at *all* points along the resilience curve.

# 7. Conclusion

While an argument could be made that, given the approximated validation results, ANN could be a viable option in modelling recovery, the authors would suggest that additional research be conducted to better understand what variables contribute to overall recovery when considering the socio-physical interactive nature of community resilience. The lack of discernible differences in the graphical analysis and case-study validation of two substantially different RMs remains a concern. Additionally, the social variables considered to have strong connections in the RMs were not the same as those from the original damage models, even though both highlight building height as an important structural variable for these modelling purposes. The graphical analysis of these ANNs appears to reinforce me of a model's applicability and of the general principle that machine learning methods are best used in situations where contributing variables are already known, even if the *how* of those variables' interactions are not.

Data accessibility. Data available from the Dryad Digital Repository: https://doi.org/10.5061/dryad.9kd51c5jb [15].
Authors' contributions. S.F.P. and H.M. conceived and designed the study. S.F.P. collected and analysed the data and wrote the initial draft of the paper and H.M. supervised the work and edited the manuscript.
Competing interests. The authors declare no competing interests.

Funding. This study was funded as part of cooperative agreement 70NANB15H044 between the National Institute of Standards and Technology (NIST) and Colorado State University and is gratefully acknowledged. The content expressed in this paper are the views of the authors and do not necessarily represent the opinions or views of CSU, NIST or the US Department of Commerce. The data used in this study will be available upon request.

Acknowledgements. The authors wish to thank the National Institute of Standards and Technology for funding this research.

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
