## [Peer Review File · Royal Society Open Science]

Review History

RSOS-211014.R0 (Original submission)

Review form: Reviewer 1

Is the manuscript scientifically sound in its present form?

Yes

Are the interpretations and conclusions justified by the results?

Yes

Is the language acceptable?

Yes

Do you have any ethical concerns with this paper?

No

Have you any concerns about statistical analyses in this paper?

No

Recommendation?

Accept with minor revision (please list in comments)

Comments to the Author(s)

This paper investigates the use of different ANN models that incorporate social variables to estimate the repair and recovery time of buildings after extreme wind events. In a companion paper published by the same authors, similar simulation tools were used to estimate the damage state of buildings post wind events. The reviewer believes that this is a novel work, nicely complementing the authors' previous research, that provides valuable information for the structural engineering community and beyond for controlling wind-related disasters and is recommended for publication.

Minor comments are as below:

- There are two tables marked as table one. Please either merge the tables together or mark with different numbers. Additionally, the first table is not cited within the manuscript. Citing the first paper in section 4 of the manuscript will help the reader better understand the differences between the recover models proposed by the authors.
- There are two tables marked as table one. Please either merge the figures together or mark with different numbers. Subsequently, figures 5 and 6 should be numbered 6 and seven both in figure captions and within the text.

Decision letter (RSOS-211014.R0)

Dear Dr Mahmoud

On behalf of the Editors, we are pleased to inform you that your Manuscript RSOS-211014 "Update Article: Applicability of Artificial Neural Networks to integrate Socio-Technical Drivers of Buildings Recovery Following Extreme Wind Events" has been accepted for publication in Royal Society Open Science subject to minor revision in accordance with the referees' reports. Please find the referees' comments along with any feedback from the Editors below my signature.

Please submit your revised manuscript and required files (see below) no later than 7 days from today's (ie 25-Oct-2021) date. Note: the ScholarOne system will 'lock' if submission of the revision is attempted 7 or more days after the deadline. If you do not think you will be able to meet this deadline please contact the editorial office immediately.

on behalf of Professor Weisi Guo (Associate Editor) and R. Kerry Rowe (Subject Editor)
openscience@royalsociety.org

Reviewer comments to Author:

Reviewer: 1

Comments to the Author(s)

This paper investigates the use of different ANN models that incorporate social variables to estimate the repair and recovery time of buildings after extreme wind events. In a companion paper published by the same authors, similar simulation tools were used to estimate the damage state of buildings post wind events. The reviewer believes that this is a novel work, nicely complementing the authors' previous research, that provides valuable information for the structural engineering community and beyond for controlling wind-related disasters and is recommended for publication.

Minor comments are as below:

- There are two tables marked as table one. Please either merge the tables together or mark with different numbers. Additionally, the first table is not cited within the manuscript. Citing the first paper in section 4 of the manuscript will help the reader better understand the differences between the recover models proposed by the authors.
- There are two tables marked as table one. Please either merge the figures together or mark with different numbers. Subsequently, figures 5 and 6 should be numbered 6 and seven both in figure captions and within the text.

===PREPARING YOUR MANUSCRIPT===

one version should clearly identify all the changes that have been made (for instance, in coloured highlight, in bold text, or tracked changes);

While not essential, it will speed up the preparation of your manuscript proof if you format your references/bibliography in Vancouver style (please see

<https://royalsociety.org/journals/authors/author-guidelines/#formatting>). You should include DOIs for as many of the references as possible.

===PREPARING YOUR REVISION IN SCHOLARONE===

<https://royalsociety.org/journals/authors/author-guidelines/#data>. You should ensure that you cite the dataset in your reference list. If you have deposited data etc in the Dryad repository,

please only include the 'For publication' link at this stage. You should remove the 'For review' link.

-- If you are requesting an article processing charge waiver, you must select the relevant waiver option (if requesting a discretionary waiver, the form should have been uploaded, see 'File upload' above).

-- If you have uploaded any electronic supplementary (ESM) files, please ensure you follow the guidance at <https://royalsociety.org/journals/authors/author-guidelines/#supplementary-material> to include a suitable title and informative caption. An example of appropriate titling and captioning may be found at https://figshare.com/articles/Table_S2_from_Is_there_a_trade-off_between_peak_performance_and_performance_breadth_across_temperatures_for_aerobic_scope_in_teleost_fishes_/3843624.

Author's Response to Decision Letter for (RSOS-211014.R0)

See Appendix A.

Decision letter (RSOS-211014.R1)

Dear Dr Mahmoud,

I am pleased to inform you that your manuscript entitled "Update Article: Applicability of Artificial Neural Networks to integrate Socio-Technical Drivers of Buildings Recovery Following Extreme Wind Events" is now accepted for publication in Royal Society Open Science.

The proof of your paper will be available for review using the Royal Society online proofing system and you will receive details of how to access this in the near future from our production office (openscience_proofs@royalsociety.org). We aim to maintain rapid times to publication after acceptance of your manuscript and we would ask you to please contact both the production office and editorial office if you are likely to be away from e-mail contact to minimise delays to

publication. If you are going to be away, please nominate a co-author (if available) to manage the proofing process, and ensure they are copied into your email to the journal.

on behalf of Professor Weisi Guo (Associate Editor) and R. Kerry Rowe (Subject Editor)
openscience@royalsociety.org

Appendix A

Update Article: Applicability of Artificial Neural Networks to integrate Socio-Technical Drivers of Buildings Recovery Following Extreme Wind Events

RESPONSE TO EDITOR:

General Comments:

===MATTERS TO ADDRESS===

===DETAILS AND COMMENTS STEP===

1 -- DATA, CODE, MATERIALS SUPPORTING YOUR PAPER --

We note that you have provided data and code within Google Drive. Before we can proceed, we require all data and code to be in a recognised research data repository, in-line with our policies: <https://royalsociety.org/journals/authors/author-guidelines/#data> At the revision stage, data and code should not be in a cloud service due to their lack of permanence.

It is an absolute condition of submission to and publication in the journal that you make available the dataset(s), code, and other digital research materials supporting the results in the article. These must be provided for our Editors and reviewers for peer-review, and then made publicly available at acceptance. You must supply sufficient data, code, and digital research materials to allow an interested reader to attempt a replication of your study.

You should provide full details of how to access the data, code, or other digital research materials in the 'Data' question (hereafter, the 'data access statement') of the ScholarOne electronic submission form at 'Step 6 – Details and Comments'. Datasets, code and other digital research materials should be deposited prior to submission in an appropriate publicly available repository and details of the associated accession number, link to and DOI of the datasets must be included in the Data Availability section of the article. See <https://royalsociety.org/journals/authors/author-guidelines/#data> and <https://royalsociety.org/journals/ethics-policies/data-sharing-mining/>.

The data access statement should include accessible URLs and accession numbers to the dataset, code or digital materials. Reference(s) to datasets should also be included in the reference list of the manuscript with DOIs (where available). You should include both the 'for review' (sometimes referred to as a 'temporary, private link') and 'for publication' versions of the URL where they differ.

In order to make it as easy as possible to comply with this policy, the Royal Society Open Science submission system is fully integrated with the Dryad data repository (<http://datadryad.org/>) and we cover the cost of submitting data to Dryad. Data submitted as electronic supplementary material

will, upon acceptance of a manuscript, be deposited at the Royal Society's figshare portal (<https://rs.figshare.com/>) free of charge. If using Dryad, you must add the temporary Dryad URL for review alongside the Dryad DOI in your data access statement in the ScholarOne submission form.

Authors Response: Thank you for the note and suggestion. We deposited the codes and data material in Dryad. The citation of the codes and data is noted below.

Pilkington, Stephanie; Mahmoud, Hussam (2021), Data related to Update Article: Applicability of Artificial Neural Networks to integrate Socio-Technical Drivers of Buildings Recovery Following Extreme Wind Events, Dryad, Dataset, <https://doi.org/10.5061/dryad.9kd51c5jb>

Please note that editors and reviewers are unable to see the data submitted to Dryad via the DOI alone, as this link only becomes publicly available upon acceptance of the manuscript.

Including the Dryad review URL allows editors and reviewers to access the data when assessing the manuscript in line with the journal's data access policies (<https://royalsocietypublishing.org/rsos/for-authors#question4>). The Dryad review URL usually follows the format below, where 'XYZ123' is replaced by a string of numerals and characters:

Authors Response: Thank you for the note and suggestion. The material can be viewed at the following link.

https://datadryad.org/stash/share/q7IUtUeDMkoZO0Z1xpBzAcbjf9Xdl_Q48nXRvF4Aic0

Please do NOT deposit your manuscript figure or table files in the Dryad repository (<http://datadryad.org/>) -- this may cause unnecessary delays. Only datasets, code, or other digital research materials not already reported in your manuscript or included as electronic supplementary information should be included in the Dryad repository.

Exceptions to our data, code, and supporting material deposition and availability requirements are possible under exceptional circumstances, but these must be agreed in advance with the editorial office.

====FILE UPLOAD STEP====

====ORIGINAL FILES====

At Step 3 'File upload' of the ScholarOne electronic submission form, you should now include the following files:

Authors Response: The version with the highlighted changes has been uploaded.

Authors Response: The clean version has been uploaded.

Additionally, upload:

Authors Response: Individual files for each figure have been uploaded.

Authors Response: Editable files for each table have been uploaded.

Authors Response: A file for all captions have been uploaded.

Authors Response: We don't have supplementary material

-- A copy of your point-by-point response to referees and Editors.

Authors Response: A file by file response has been included

Please note that, if you are requesting a discretionary waiver for the article processing charge, the waiver form must be included at this step. Please see <https://royalsocietypublishing.org/rsos/waivers#question4>.

Please add these files and then resubmit the paper for consideration.